# Metamaterial assisted illumination nanoscopy via random super-resolution speckles

Yeon Ui Lee [1,5], Junxiang Zhao[1,5], Qian Ma[1,5], Larousse Khosravi Khorashad [1], Clara Posner [2], Guangru Li[1], G. Bimananda M. Wisna[3], Zachary Burns[1], Jin Zhang [2] & Zhaowei Liu [1,3,4 ✉]

Structured illumination microscopy (SIM) is one of the most powerful and versatile optical super-resolution techniques. Compared with other super-resolution methods, SIM has shown its unique advantages in wide-field imaging with high temporal resolution and low photon damage. However, traditional SIM only has about 2 times spatial resolution improvement compared to the diffraction limit. In this work, we propose and experimentally demonstrate an easily-implemented, low-cost method to extend the resolution of SIM, named speckle metamaterial-assisted illumination nanoscopy (speckle-MAIN). A metamaterial structure is introduced to generate speckle-like sub-diffraction-limit illumination patterns in the near field with improved spatial frequency. Such patterns, similar to traditional SIM, are then used to excite objects on top of the surface. We demonstrate that speckle-MAIN can bring the resolution down to 40 nm and beyond. Speckle-MAIN represents a new route for super-resolution, which may lead to important applications in bio-imaging and surface characterization.

[1] Department of Electrical and Computer Engineering, University of California, San Diego, CA 92093, USA. [2] Department of Pharmacology, University of California San Diego, San Diego, CA 92093, USA. [3] Material Science and Engineering Program, University of California, San Diego, CA 92093, USA. [4] Center for Memory and Recording Research, University of California, San Diego, CA 92093, USA. [5]These authors contributed equally: Yeon Ui Lee, Junxiang Zhao, Qian Ma. ✉email: zhaowei@ucsd.edu

Structured illumination microscopy (SIM)[1] is a widefield technique in which a series of illumination patterns are generated and superimposed on to a specimen while capturing images. Images beyond the diffraction limit can then be obtained by using a reconstruction algorithm. Compared with other super-resolution approaches such as single molecule localization based methods[2], like photoactivated localization microscopy (PALM)[3] and stochastic optical reconstruction microscopy (STORM)[4]; or reversible saturable optical linear fluorescence transitions microscopy[5], such as stimulated emission depletion[6] and ground state depletion microscopy[7], SIM has been proven to have supreme advantages in high spatial-temporal resolution associated with low photo-toxicity[8,9], and thus has recently attracted more attention.

The resolution of traditional SIM, however, is limited by its highest attainable spatial frequency $f$ contributed from both illumination and detection:

$$f = f_{det} + f_{illum} \tag{1}$$

where $f_{det}$ and $f_{illum}$ are the maximum spatial frequencies of the detection optics and illumination patterns, respectively. By using far-field optics, both illumination and detection are diffraction-limited. Thus, traditional SIM only extends the resolution by a factor of ~2 compared to the diffraction limit (the wavelength difference between illumination and detection is ignored), representing the major bottleneck of this promising technology.

To further improve the resolution, researchers explored various ways to extend the spatial-frequency $f_{illum}$ of illumination patterns. For instance, nonlinear SIM[10], by saturated excitation of fluorophores, can utilize higher order harmonics and access much larger $f_{illum}$. However, it requires much stronger laser intensity, which is inevitably associated with significant phototoxicity and photobleaching. Recently, sub-60-nm resolution was achieved by combining total internal reflection fluorescence (TIRF)-SIM/lattice light sheet microscopy and nonlinear SIM based on the patterned activation of a reversibly photoswitchable fluorescent protein (PA NL-SIM)[9] at subsecond acquisition speeds over hundreds of time points in multiple color near the basal plasma membrane.

The resolution of illumination patterns can also be increased by the substrate material if the specimen is illuminated by an evanescent wave. In this case, only specimens on the substrate surface can be imaged. For instance, TIRF-based SIM can take advantage of the high refractive indices ($n$) of its substrates (Glass[11] $n$ = 1.46–1.52; $Al_2O_3$[12] $n$ = 1.77; GaP[13,14] $n$ = 3.6), generating an illumination pattern with smaller features. Plasmonic materials provide an alternative approach to create super-resolution illumination patterns beyond the traditional limit. For instance, plasmonic structured illumination microscopy (PSIM[15–17]), utilizing propagating surface plasmon waves to form interference patterns, has shown ~2.5-fold resolution improvement to the diffraction limit; localized plasmonic structured illumination microscopy (LPSIM[18]) further extends the resolution by utilizing a substrate consisting of an array of metallic nanodisks that generates angle-sensitive localized surface plasmons as illumination sources. LPSIM with ~3-fold resolution improvement to the diffraction limit has recently been demonstrated in experiments[19]. Hyperbolic metamaterials have also been introduced to SIM to further extend the resolution of illumination patterns[20,21], but has only been theoretically studied.

In this work, we introduce a speckle metamaterial-assisted illumination nanoscopy (speckle-MAIN) technique, which has the potential to extend the SIM microscopy resolution to deep subwavelength scales. A widely used multilayer hyperbolic metamaterial (HMM) is employed to carry high-spatial-frequency waves[20,21], and also form random speckles as ultra-high-resolution illumination patterns. Benefiting from those high-resolution illumination patterns, the proof-of-concept experiment showed that speckle-MAIN can resolve two-beads with center-to-center resolution of ~40 nm while remaining a relatively simple, mass-producible sample preparation method. Because the illumination pattern is mainly enabled by the HMM substrate, it can greatly reduce the complexity of the optical system. We acquire images by a slightly modified commercial inverted microscope and retrieve super-resolution by the blind structured illumination method[22], which was developed for diffraction-limited speckle illuminations.

## Results

Hyperbolic metamaterials, benefiting from their anisotropic hyperbolic isofrequency curve, are capable of carrying much higher spatial frequency contents than most materials that can be found in nature. We show the comparison among isofrequency curves of air, an 'ideal' HMM and a practical HMM multilayer in Fig. 1a. The ideal HMM has an unlimited $k$-space, while a practical HMM, made by Ag-$SiO_2$ multilayer, has a $k$-space limitation from its periodicity[23–26]. A multilayer with a smaller period of alternating layers supports a higher spatial frequency. A practically achievable Ag/$SiO_2$ multilayer with a period of 20 nm, supports a highest $k$ mode around $10k_0$ working at 488 nm wavelength[23]. Speckle patterns, as a result of multi-beam interference of the waves through the HMM, will have their resolution enhanced by the enlarged $k$ bandwidth, as shown in Fig. 1b. We take advantage of intrinsic surface scattering (see Fig. 1d) as well as volumetric scattering caused by non-perfect multilayers to convert the incident plane wave to high $k$ vector waves and generate ultrahigh resolution speckle on the top surface. Compared to the subwavelength near-field speckle patterns generated by using random nanoparticles[27,28], the HMM assisted speckles possess much higher resolution. The high-resolution speckle patterns can also be translated or modified by changing the incident angles and wavelengths as shown in Fig. 1c. In the following, we firstly characterize the optical property and morphology of the Ag-$SiO_2$ multilayer HMM made by sputtering deposition method. Second, we simulate the effect of non-flat interfaces in generating the near field illumination speckles. Lastly, we demonstrate the super resolution capability of the proposed speckle-MAIN by imaging various specimen including fluorescently labeled cells.

The HMM structure consists of 3 pairs of Ag and $SiO_2$ layers. This multilayer is deposited on a glass substrate by sputtering (Supplementary Fig. 1). Figure 2 presents the characterization results of this HMM. Optical transmission/reflection of the multilayer at normal incidence is measured and then compared to the theoretical calculation using bulk material property (Fig. 2a). We measured that, when the thickness of the silver film is below 10 nm, the transmission of the HMM drastically deviates from the calculated one, indicating the continuous silver films start to break. We select 10 nm for the silver film to ensure the accuracy of our simulation by using bulky silver material properties in the following sections. Atomic force microscopy (AFM) is used to measure the top morphology of the deposited multilayer (Fig. 2b). The roughness (RMS) of the multilayer is ~1.1 nm, and the correlation length of the roughness is 35 nm. The transmission electron microscopy (TEM) image (Fig. 2c) presents the well-formed continuous layers in the HMM. The non-perfect interface of these two amorphous thin films can also be observed.

To foresee the effect of small roughness on generating speckles, a full wave simulation modeling with measured film roughness is used (see methods for more details). We excite the HMM using $x$-polarized plane wave with wavelength $\lambda$ and varying polar angle $\theta$. The speckles,

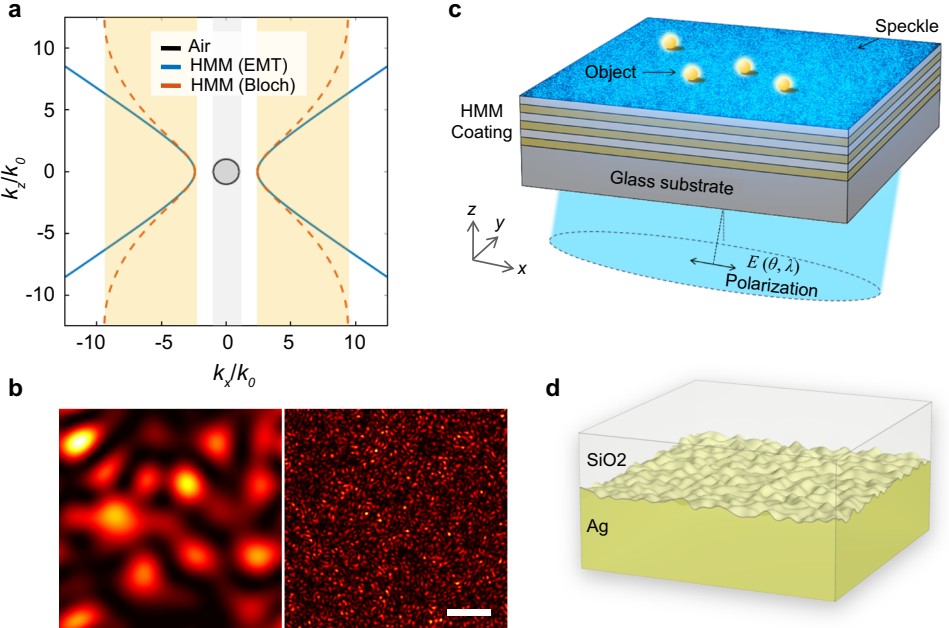

**Fig. 1 Metamaterial-assisted speckle illumination. a** Isofrequency curve of air, an ideal HMM by effective medium theory (EMT), and a practical HMM consists of periodical layered structures of Ag and $SiO_2$ (Bloch) at wavelength of 488 nm. The wave vector $k_x$ and $k_z$ are both normalized to the wave vector in air $k_0$. The allowed $k$-bandwidth is highlighted in gray (air) and in orange (practical HMM). **b** Calculated exemplary speckle patterns (normalized intensity) from different material systems. Left: diffraction limited speckle (gray bandwidth in (**a**)); Right: HMM assisted speckle (yellow bandwidth in (**a**)). Scale bar: 400 nm. **c** A HMM coated substrate projects ultra-fine structured speckles onto objects lying on its top surface. **d** Schematics of the non-uniform interface between sputtered Ag and $SiO_2$ interface.

represented by the field intensity distribution 10 nm away from the top surface of HMM, are simulated with a series of wavelengths and incident angles. Figure 2(d–f) present three selected $|E|^2$ distributions at (500 nm, 0°); (500 nm, 45°) and (600 nm, 0°), respectively, demonstrating that the speckle has ultrahigh resolution and can be controlled by tuning either wavelength or incident angle/phase.

Two parameters of the speckle patterns are highly relevant to the SIM application: the spatial resolution of the speckles and the independence between different speckles. The first one is analyzed based on the magnitude transfer function (MTF) of the $|E|^2$ (Supplementary Fig. 2). Generally, a multilayer HMM with thinner unit pair size will have speckles with higher resolution. However, the unit pair size is limited by the minimum silver film thickness below which silver will become isolated islands instead of continuous films. The latter can be analyzed by the cross-correlation between the speckles (Supplementary Fig. 3). The correlation between speckles drops to 0.5 with ~15-degree incident angles difference when the operating wavelength is 488 nm.

Considering the illuminating system is a linear system with a complex electric field, the intensity distribution can be altered from the phase difference between beams that have different incident angles. To have enough distinguishable speckles, we illuminate the HMM with a random, diffraction-limited optical field generated by either a diffuser or a multimode fiber. The complex fields (diffraction-limited speckles), equal to a composite of plane waves at different angles and phases, will be converted into sub-diffraction-limited speckles after passing through the HMM. At the sample plane, the high-resolution speckles excite the fluorophores in a specimen. The fluorescence signal is then directly collected by a standard inverted microscope system (Fig. 2g). Note that compared to a conventional structured illumination microscopy, speckle-MAIN can have comparable imaging speed per sub-frame. Illumination efficiency through HMM is about 15–20% at 488 nm, as shown in Fig. 2a, which can be compensated by increasing laser power.

Since speckle-MAIN utilizes high resolution near field excitation for super-resolution, the excitation field intensity decreases exponentially with distance from the HMM. The supported large wavevector speckle illumination patterns are sensitive to the distance to the HMM surface (Supplementary Fig. 8). In general, the closer an object resides to the HMM surface, the higher maximum resolution can be achieved as more high-$k$ components of the illumination patterns will be utilized (see Supplementary Fig. 8 and Supplementary Fig. 9 for details).

To demonstrate the super-resolution capability of the proposed speckle-MAIN experimentally, we image sparsely distributed fluorescent beads (Fluoresbrite YG Carboxylate, ~46 nm in diameter) drop-casted on the HMM-coated glass substrate. A 488 nm CW laser, directed by a multi-mode fiber, is incident on the other side of the HMM substrate. The incident optical field is scrambled due to the multi-mode interference and can be changed by adding vibration to the fiber (Fig. 2g, see methods and Supplementary Fig. 4 for more details on experimental set up). The incident field has a numerical aperture (NA) of ~0.2. Based on the simulation, the illumination pattern on the fluorescent beads will have much higher resolution after passing through the HMM.

Figure 3a and c presents a few selected camera frames while incident optical field is changed randomly by the vibrating multimode fiber. The video is taken under a $100 \times /1.5$ oil objective lens to maximize the detection NA. Knowing that the sample is made of fluorescent spheres, the unsymmetrical look of the diffraction-limited image indicates the presence of multiple beads. Four selected raw camera frames (Fig. 3c) present obvious peak location shifts during the acquisition, indicating the changes of illumination patterns at high-resolution. The super-resolution image (Fig. 3b right panel) is reconstructed by using blind-SIM algorithms from all 500 frames (see methods for details). The super-resolution image resolves four well separated beads, with the closest two-bead spacing to be 80 nm. The location of the

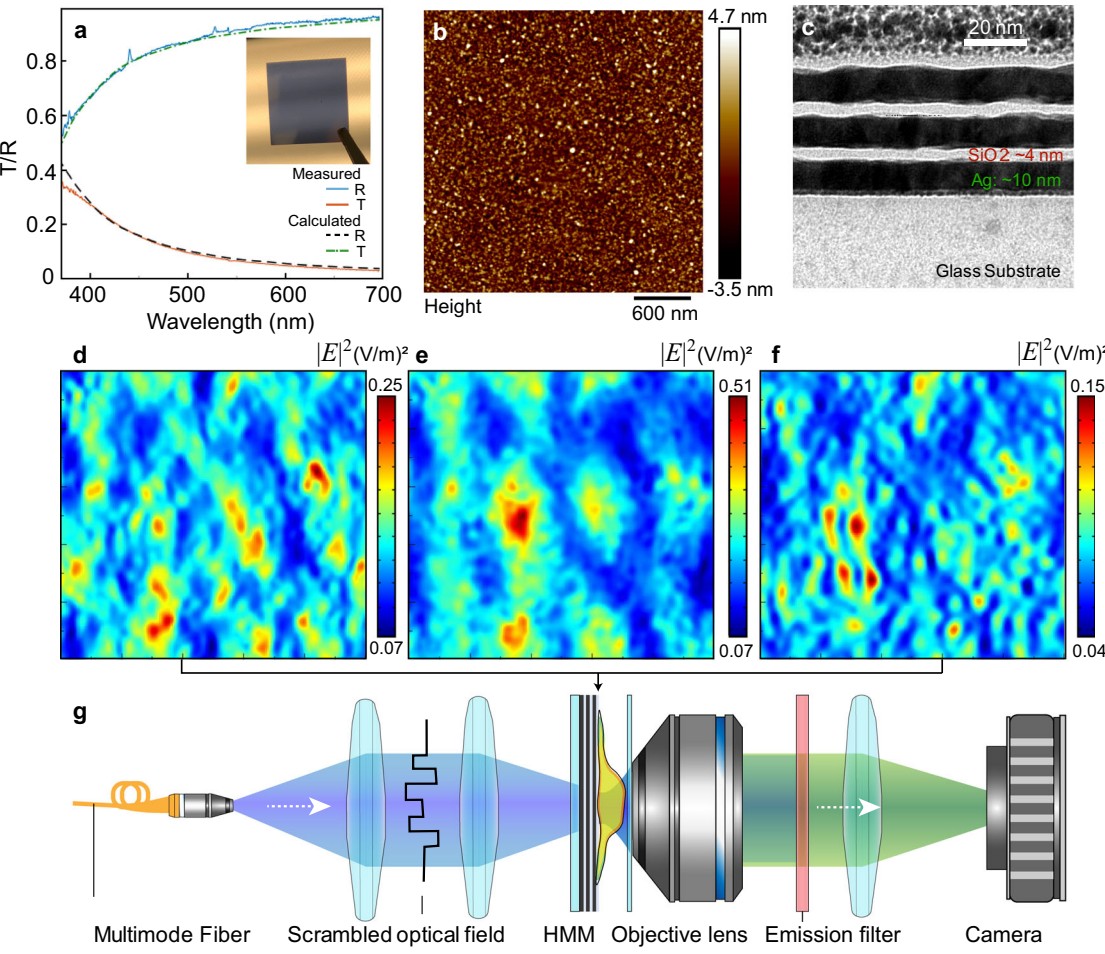

**Fig. 2 Multilayer HMM characterization and simulation. a** Transmittance (T) and reflectance (R) at normal incidence of the Ag-SiO$_2$ multilayer HMM. Inset image: a photo of the HMM-coated cover glass. Calculation is done based on transfer matrix method for 3 pairs of {10 nm Ag and 4 nm SiO$_2$} on glass. **b** AFM image of the top surface. Scan area: 3 μm × 3 μm (512 pixels × 512 pixels). **c** TEM cross-section shows 3 pairs of 10 nm Ag and 4 nm SiO$_2$ on top of a glass substrate with Cr adhesion layer. **d–f** Simulated light intensities ($|E|^2$) on x-y plane at 10 nm on top of the HMM. The total simulated size of illumination pattern is 1 μm × 1 μm. **d** $\theta = 0°$, $\lambda = 500$ nm; e, $\theta = 45°$, $\lambda = 500$ nm; **f** $\theta = 0°$, $\lambda = 600$ nm. $\theta$ is the incident angle; $\lambda$ is the operating wavelength. **g** Schematics of the experimental set-up of speckle-MAIN.

reconstructed four beads are marked on the raw frames, which agrees well with the peak intensity locations of those frames. Considering the low resolution of the far field incident light (NA ~0.2), the sub-diffraction changes of illuminating pattern is primarily caused by the HMM substrate. In a separate location shown in Fig. 3d, a diffraction-limited spot is shown to consist of two touching fluorescent beads after the reconstruction. The center-to-center separation is measured to be 40 nm, demonstrating the speckle-MAIN's ultrahigh resolution capability. We introduce an image-resolution measure, i.e., Fourier ring correlation (FRC)[29] that can be computed from experimental data (Supplementary Fig. 11). The standard 1/7 FRC resolution criteria illustrates 43 nm Fourier space cutoff with speckle-MAIN.

From an information theory point of view, at least $N^2$ sub-frames are needed to reconstruct a super resolution image with $N$-fold resolution improvement. Traditional SIM utilizes well known sinusoidal patterns to sample the object with high efficiency, so that a small number, i.e., close to $\alpha N^2$, of sub-frames are involved. $\alpha$ is the oversampling factor. Speckle-based blind-SIM, however, requires more frames owing to the lack of knowledge regarding exact illumination patterns and also the correlations between the illumination patterns. After reduction to only 80 sub-frames, we still can resolve two particles with ~60 nm center-to-center distance by using an objective with NA = 0.8

(Fig. 4a–i), indicating the robustness of the speckle-MAIN technology.

Figure 4j–m represents a super-resolution image of a more complex object at a wide field of view. The object is made by a dense drop-casting of quantum dots emitting at 605 nm. The quantum dots form certain nanostructures on top of the HMM-coated substrate. The super-resolution image reconstructed from 140 sub-frames reveals much greater details that cannot be observed by diffraction-limited images. In principle, speckle-MAIN has a field of view (FOV) as wide as an epi-fluorescent microscope, which is primarily limited by the camera sensor size. The standard 1/7 FRC resolution criteria (Supplementary Fig. 11) illustrates 65 nm Fourier space cutoff with speckle-MAIN.

Next we test the applicability of speckle-MAIN with biological samples since bioimaging is one of the most important applications for super-resolution microscopy. We deposit a thin SiO$_2$ protection layer (~10 nm) with minimal defects on top of HMM before cell growth to not only increase the biocompatibility of the HMM substrate but also reduce the quenching effect of the fluorescence dyes. Cos-7 cells transiently transfected with fluorescently labeled actin-binding Lifeact (Lifeact-Venus) are fixed and subjected to the speckle-MAIN measurement (see Fig. 5a). Upon excitation with a 488 nm laser, we acquire 500 frames (1 frame per second) by changing the scrambled incident optical

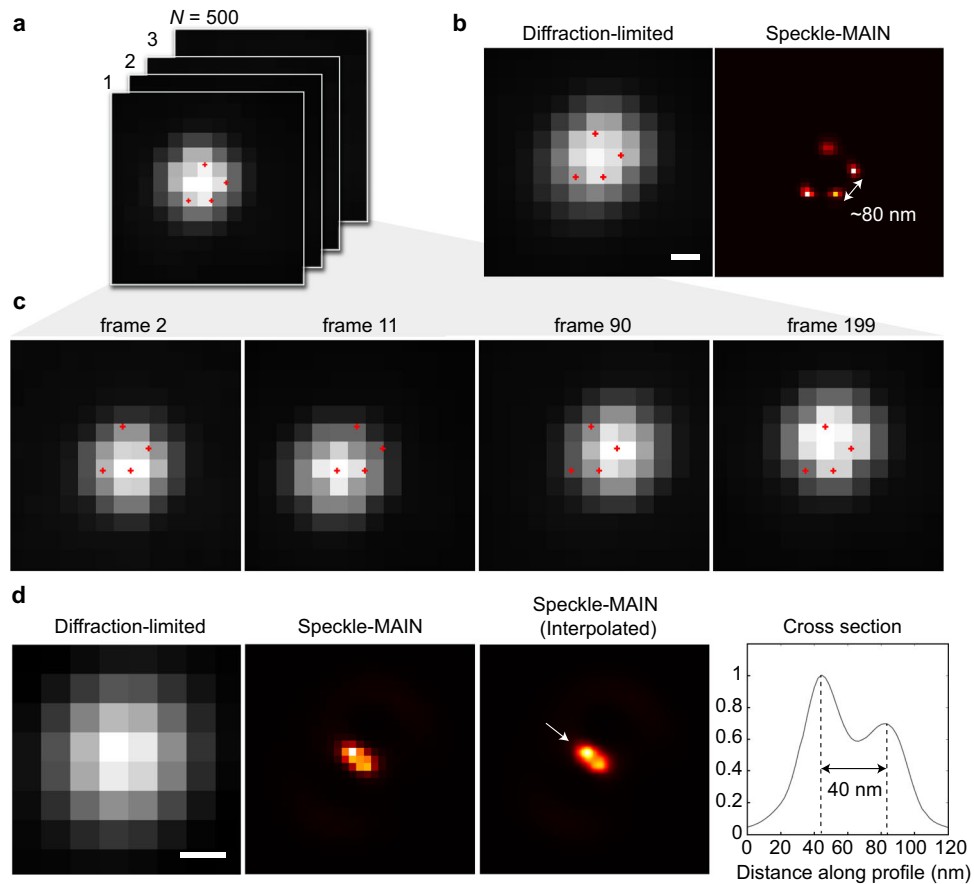

**Fig. 3 Super resolution speckle-MAIN demonstration. a** An intensity normalized image stack of fluorescent beads on HMM-coated substrate with different illumination condition. Objective lens: Olympus 100×/1.5 oil. Frame rate: 1 Hz. Exposure time: 500 ms. Total frame: 500. **b** Diffraction-limited image and the reconstructed speckle-MAIN image. The speckle-MAIN image is reconstructed from 500 diffraction-limited frames. Pixel size: diffraction-limited, 65 nm; speckle-MAIN, 22 nm. Locations of the reconstructed beads are marked in those raw frames. The diffraction-limited image is computed by averaging all 500 frames. Scale Bar: 100 nm. **c** Four selected frames indicate a sub-diffraction changes of illumination pattern. **d** Imaging of two close fluorescent beads shows speckle-MAIN resolution down to ~40 nm. The cross section is plotted along the arrow direction in the interpolated speckle-MAIN image. Speckle-MAIN (interpolated) has 10 times interpolation from speckle-MAIN image. Pixel size: diffraction-limited, 65 nm; speckle-MAIN, 16 nm. Scale bar: 100 nm.

field. We use 200 ms exposure time and 1 Hz frame rate to wait for the multimode fiber to stabilize in Fig. 5. The fluorescent signal is collected by an objective lens (40× /0.6 NA Olympus objective) with proper emission filter (520/40 nm band-pass filter). A set of 200 frames are used for image reconstruction. Because all the metallic structures are embedded in glass, all the cell preparation processes for speckle-MAIN are identical to the case of using a conventional glass slide. The wide-field of view image reconstruction takes 10 mins on a desktop computer with a GTX 1080Ti graphics card and a i7-8700k CPU to reconstruct an image with 100 by 100 raw pixels. Figure 5c, d and e, f presents zoomed in images in the indicated regions for the diffraction-limited image and speckle-MAIN reconstruction correspondingly. In the reconstructed image, the actin filaments are well resolved with fine features, which are not clearly discernible in the diffraction-limited image. We also find some fragmented features, which might be induced by the interaction between the HMM substrate and fluorophores and/or substrate-dependent cell attachment. This example indicates that the proposed speckle-MAIN technique can be directly applied to cell imaging without any modifications of the existing sample preparation protocols. Future efforts will explore more biocompatible HMMs so that the protection layer could be eliminated for better imaging performance.

In conclusion, we propose and demonstrate speckle-MAIN, a robust, wide-field and low-cost super-resolution method with resolution down to ~40 nm. The microscope set up is greatly simplified compared to other structured illumination methods or engineered point spread function techniques. Moreover, speckle-MAIN does not rely on any specific properties from fluorescent molecules, so that any fluorescent probes can be used. The experiment can be conducted under a conventional epi-fluorescent microscope, with an HMM-coated substrate and a multi-mode fiber for illumination. Compared to traditional SIM, speckle-MAIN greatly extends the spatial resolution with a cost of increased number of measurements. Compared to localization techniques like PALM/STORM, speckle-MAIN, like other structured illumination-based technology, has the advantages of low incident power, and a smaller number of frames. Note that speckle-MAIN does not solely rely on the blind-SIM reconstruction and several other reconstruction methods such as SOFI[30,31], ESI[32,33], and MUSICAL[34] can also be used to further improve the image quality, imaging speed and resolution. Furthermore, additional prior information of the object can be combined with the pre-determined illumination information to increase the imaging speed dramatically. Contrary to other fluctuation based super-resolution techniques, the generated super-resolution speckle patterns in speckle-MAIN can be replicated

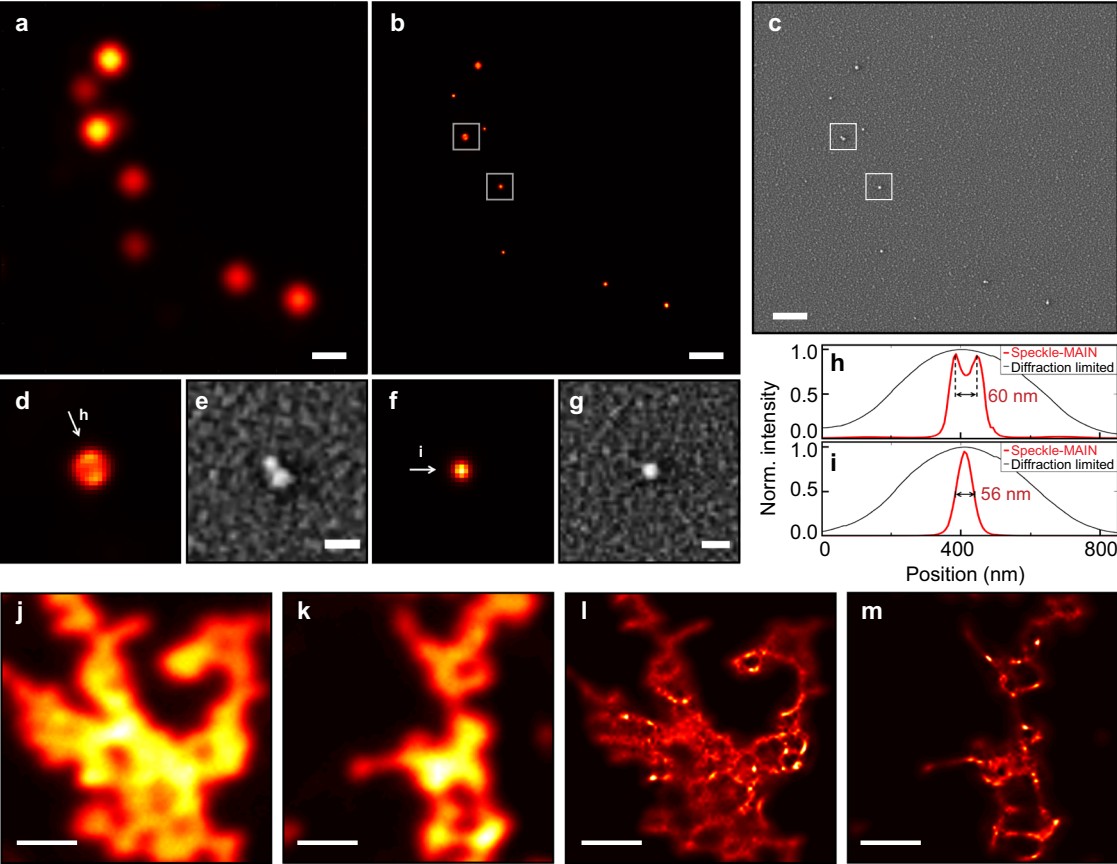

**Fig. 4 Speckle-MAIN super-resolution image of fluorescent beads and Qdot 605. a** Diffraction-limited image of fluorescent beads. **b** Reconstructed speckle-MAIN image. **c** SEM image. Scale bar: 600 nm. **d–g** zoomed-in images (**b**) and (**c**) of fluorescent beads. Scale bar: 100 nm. **h, i** Normalized intensity cross-section (red curves) of images (**d**, **f**) along indicated direction. Black curves show the corresponding intensity cross-section of conventional wide-field images. **j, k** Diffraction-limited image. Objective lens: 50×/0.8 NA. **l, m** Reconstructed speckle-MAIN image of Qdot 605. Scale bar: 2 µm. Exposure time: 200 ms, frame rate: 1 fps.

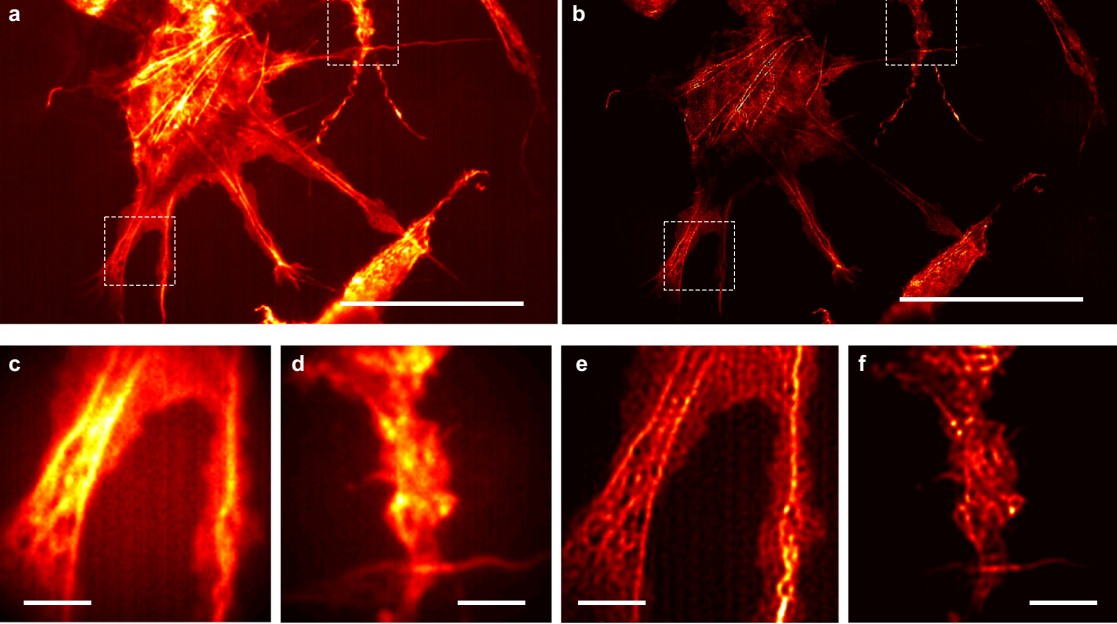

**Fig. 5 Speckle-MAIN imaging of Cos-7 cells. a** Diffraction-limited image. Scale bar: 20 µm. Objective lens: 40 ×/0.6 NA. **b** Reconstructed speckle-MAIN image. **c, d** Zoom-in view of the white box area in (**a**). **e, f** Zoom-in view of the white box area in (**b**). Scale bar: 2 µm.

under the exact same illumination condition. The deterministic nature of the speckle enables more prior information for image reconstruction given the known nanostructures in the HMM substrate. The resolution of speckle-MAIN is mainly limited by the metamaterial itself, i.e., the unit pair size of the HMM multilayers as well as the quality of the interfaces and volumetric scatters. Therefore, a breakthrough in HMM manufacturing may lead significant resolution improvement for the speckle-MAIN technology.

## Methods

**Simulation of non-flat multilayer HMM**. The results of the simulations presented here are based upon the solutions of coupled Maxwell's equations. Finite difference time domain method was employed using Lumerical software. In the simulation, it is sufficient to excite the HMM using a x-polarized electromagnetic plane wave with wavelength $\lambda$ and varying polar angle $\theta$ in order to observe the speckle pattern changes. We incorporated local bulk dielectric functions of Ag from literature[35]. The refractive index of $SiO_2$ was chosen to be equal to 1.46. The background was chosen as air like in experiment. To consider the non-flat or random rough surface, it is important to model the rough surfaces accurately to describe the main characteristics observed in our sample. Topographic features of a random rough surface are characterized mainly by a root mean square (RMS) as a measure of the magnitude of varying height and correlation length ($C_l$) in two dimensions where the surface is defined[36]. Based upon our measurement (Supplementary Figs. 6 and 7), we chose RMS = 1.1 nm and $C_l$ = 35 nm to model the random rough surfaces (see Supplementary Fig. 7 for the comparison between the code-generated rough surface and measured ones). After the multilayer HMM is excited by the incident light, the speckle intensity distribution is calculated at 10 nm above the top surface of the structure.

**Experiment setup**. A 488 nm laser (Coherent Genesis MX488-1000 STM) is coupled into a multimode fiber (Thorlabs, core diameter: 50 μm, NA 0.22). The other end of the fiber is coupled to a reflective fiber collimator that is attached to a custom-made adapter of the microscope condenser (Supplementary Fig. 4). The fiber end is imaged to the multilayer HMM, projecting diffraction-limited speckle patterns. The typical delivered laser intensity onto the multilayer HMM is around $10^3$ W/cm². The HMM converts the pattern into high-resolution speckles that illuminates the object on the other side. The HMM-coated substrate has transmission of ~15% at 488 nm and ~30% at 405 nm. The speckle is controlled by a step motor which will stretch the fiber spool during image acquisition. For Fig. 4, an additional ×2 magnification is used at the image plane. A sCMOS camera is used for imaging acquisition (Hamamatsu Orca Flash 4.0 v3).

**Imaging reconstruction**. All of our image processing and reconstruction are performed in MATLAB. The iterative reconstruction algorithm blind-SIM[22], which does not require exact knowledge of the illumination pattern, is used to retrieve the object information. For blind-SIM, an assumption is made that all illumination patterns add up to a uniform pattern. Both the object and the illumination patterns are treated as unknowns in real space and are solved using a cost-minimization strategy. Each super-resolution frame is reconstructed from multiple sub-frames (80–500 frames) that were illuminated under different speckle patterns. The GPU based reconstruction for an image size of 200 × 200 pixels typically takes 10–30 min on a Nvidia GTX 1080 Ti.

**Cell culture and transfection**. Cos7 (ATCC® CRL-1651) cells were cultured in Dulbecco modified Eagle medium (DMEM; Gibco) containing 4.5 g/L glucose and supplemented with 10% (v/v) fetal bovine serum (FBS, Sigma) and 1% (v/v) penicillin-streptomycin (Pen-Strep, Sigma-Aldrich). All cells were maintained in a humidified incubator at 37 ℃ with a 5% $CO_2$ atmosphere. 24 h prior to transfection, cells were seeded onto HMM substrate and grown to 50 – 70% confluence. Cells were then transfected with 100 ng of pcDNA3-Lifeact-Venus (Addgene plasmid # 87613) using Lipofectamine 2000 (Invitrogen) and grown an additional 24 h before fixation. Cells were washed with Phosphate-buffered saline (PBS) before fixation with 4% paraformaldehyde and 0.2% glutaraldehyde PBS for 10 min at room temperature. Cells were quickly rinsed in PBS after fixation and quenched with freshly made 0.1% NaBH4 ice-cold PBS. After quenching, cells were washed three times for five minutes each with PBS on a shaker. Cells were imaged at room temperature and stored in PBS at 4 ℃.

**Reporting summary**. Further information on research design is available in the Nature Research Reporting Summary linked to this article.

## Data availability

The raw image files used in this paper are available on https://www.zliugroup.com/data/Public/Blind_SIM_packages.html. All other data that support the findings of this study are available from the corresponding author upon reasonable request.

## Code availability

The custom-written Matlab code for speckle-MAIN reconstruction is available on https://www.zliugroup.com/data/Public/Blind_SIM_packages.html.

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

## Acknowledgements

We thank A. Bezryadina, Y. Han for discussion of microscope set up, Q. Yang for the AFM measurement of multilayer samples used in this manuscript. This work was supported by the Gordon and Betty Moore Foundation (to Z.L.) and NIH R35 CA197622 (to J.Z.). C.P. was supported by the Molecular Biophysics Training Grant, NIH Grant T32 GM008326. G.B.M.W. acknowledges two years scholarship from LPDP (Indonesia Endowment Fund for Education) for pursuing master degree in United States. This work was performed in part at the San Diego Nanotechnology Infrastructure (SDNI) of University of California, San Diego (UCSD), a member of the National Nanotechnology Coordinated Infrastructure (NNCI), which is supported by the National Science Foundation (Grant ECCS-1542148)

## Author contributions

Y.U.L., J.Z., and Q.M. contributed equally to this work under the supervision of Z.L. Y.U.L., J.Z., Q.M., and Z.L. conceived and designed the experiment. Y.U.L., J.Z., Q.M., and Z.B. performed the experiments. L.K.K., Y.U.L., and Q.M. performed the simulations. C.P. prepared biological samples under the supervision of J.Zhang. Q.M. and G.L. fabricated HMM samples. G.B.M.W. performed SEM. Y.U.L., J.Z., and Q.M. reconstructed the images, analyzed the data, and created the figures. Y.U.L., J.Z., and Q.M. wrote the paper which was revised by all authors.

## Competing interests

The authors declare no competing interests.
