## [Peer Review File · Nature Communications]

REVIEWER COMMENTS

Reviewer #1 (Remarks to the Author):

The authors have significantly revised this manuscript for resubmission to Nature communications compared to the previous version. They have added significant new content, especially regarding details about the depth penetration and the applicability of the technique to different experimental aims. They have also made significantly more effort to properly characterise the resolution they do achieve, including with different reconstruction approaches.

Overall the paper has been significantly improved and they have clearly answered all of the points that I originally raised. I feel that the paper is now of a sufficient standard to be published in Nature Communications.

Reviewer #2 (Remarks to the Author):

I have reviewed this manuscript twice now (Reviewer 2), and although the revised paper has additional details, on the conceptual side, there is limited novelty to grant publication in Nature Communication. The main idea of the manuscript of generating super-resolved speckle patterns and to exploit them for super-resolution microscopy is well established in the nanoscopy community.

The author in their response letter wrote "the previously established speckle illumination super-resolution methods are all limited by diffraction-limited speckles" which is incorrect.

For example, the manuscript published in 2017 in Nature Photonics, (Diekmann et. al.) clearly shows the presence of super-resolved speckle in their manuscript, see Supplementary Figure 11 of Diekmann et.al. where super-resolved speckles are demonstrated.

Diekmann, R., Helle, Ø., Øie, C. et al. Chip-based wide field-of-view nanoscopy. Nature Photon 11, 322–328 (2017). <https://doi.org/10.1038/nphoton.2017.55>

Moreover, authors commented that previous work requires specific fluorophore dyes which is also in-correct. For intensity fluctuation method to work, either intrinsic photokinetics of the fluorophores are needed or the fluctuations (modulation) of the illumination light itself is sufficient. Both the cases are well known and are being exploited by the community. In this manuscript, the authors have exploited the fluctuations caused by the illumination, itself, similar to several previous literatures, some of them copied below and many others exists.

1. Ventalon, C. & Mertz, J. Quasi-confocal fluorescence sectioning with dynamic speckle illumination. Opt. Lett. 30, 3350–3352 (2005).
2. Kim, M., Park, C., Rodriguez, C., Park, Y. & Cho, Y. H. Superresolution imaging with optical fluctuation using speckle patterns illumination. Sci. Rep. 5, 16525 (2015).
3. K. Guo, Z. Zhang, S. Jiang, J. Liao, J. Zhong, Y. C. Eldar, and G. Zheng, "13-fold resolution gain through turbid layer via translated unknown speckle illumination," Biomed. Opt. Express 9, 260–274 (2018).

The key advantage of SIM, is the "structured illumination", and by knowing the illumination periodicity, an important prior knowledge in SIM, it enables SIM with only 9/15 images for 2D/3D SIM image. If 100 images are needed, intensity fluctuations methods would also perform, which will generate results similar as shown in this manuscript (e.g. previous literature, on MUSICAL where 50 nm resolution has already being documented)

Agarwal, K., Macháň, R. Multiple signal classification algorithm for super-resolution fluorescence microscopy. *Nat Commun* 7, 13752 (2016). <https://doi.org/10.1038/ncomms13752>

On using the speckle pattern for illumination, there is no prior information and thus loss of temporal resolution becomes evident. Author stated about pre-determined illuminations using speckle idea but no scientific information was provided, so it is difficult to foresee how this would work.

The study carried out in this manuscript is systematic and is more suitable for publication in *Scientific Reports*.

Reviewer #3 (Remarks to the Author):

From my point of view the authors have addressed most of my initial concerns and the paper is now publishable in *Nature Communications*.

Before publication, I highly recommend the following changes to be made to the manuscript, though:

1) Fig. S5 should be moved to the main text - and possibly be integrated with another, already existing figure in the main text (my suggestion would be to combine it with Fig 4 in the main text). The main reason for recommending this change is that the manuscript will greatly benefit from a true proof-of-concept, which is a comparison of the reconstruction of a sub-wavelength structure with an electron micrograph of the same structure - and this is what Fig. S5 provides. It will get lost in the supplement, though....

2) The references in the supplemental information file need to be changed. Reference 2 is not an appropriate reference for SOFI. Here, the original SOFI paper by Enderlein should be cited:

Fast, background-free, 3D super-resolution optical fluctuation imaging (SOFI)

T. Dertinger, R. Colyer, G. Iyer, S. Weiss, J. Enderlein

Proceedings of the National Academy of Sciences Dec 2009, 106 (52) 22287-22292; DOI: 10.1073/pnas.0907866106

Similarly, ref. 3 is not an appropriate reference for ESI. Here, also, the original article should be cited:

Entropy-Based Super-Resolution Imaging (ESI): From Disorder to Fine Detail

I. Yahiatene, S. Hennig, M. Muller, and T. Huser

ACS Photonics 2015, 2, 8, 1049-1056

Publication Date: July 14, 2015

<https://doi.org/10.1021/acsp Photonics.5b00307>

Color codes used in this response letter:

Blue Italic: original review comments;

Black: our responses;

Red: revisions made in the manuscript.

Response for Reviewer #1

The authors have significantly revised this manuscript for resubmission to Nature communications compared to the previous version. They have added significant new content, especially regarding details about the depth penetration and the applicability of the technique to different experimental aims. They have also made significantly more effort to properly characterise the resolution they do achieve, including with different reconstruction approaches.

Overall the paper has been significantly improved and they have clearly answered all of the points that I originally raised. I feel that the paper is now of a sufficient standard to be published in Nature Communications.

[Reply] We appreciate the reviewer's positive evaluation on our study and valuable comments.

Response for Reviewer #2

[Question 2-1] *I have reviewed this manuscript twice now (Reviewer 2), and although the revised paper has additional details, on the conceptual side, there is limited novelty to grant publication in Nature Communication. **The main idea of the manuscript of generating super-resolved speckle patterns and to exploit them for super-resolution microscopy is well established in the nanoscopy community.***

The author in their response letter wrote "the previously established speckle illumination super-resolution methods are all limited by diffraction-limited speckles" which is incorrect.

For example, the manuscript published in 2017 in Nature Photonics, (Diekmann et. al.) clearly shows the presence of super-resolved speckle in their manuscript, see Supplementary Figure 11 of Diekmann et.al. where super-resolved speckles are demonstrated. Diekmann, R., Helle, Ø., Øie, C. et al. Chip-based wide field-of-view nanoscopy. Nature Photon 11, 322–328 (2017). <https://doi.org/10.1038/nphoton.2017.55>

[Answer 2-1] Discussion about novelty and super-resolved speckle patterns

We thank for the reviewer's time and efforts that have gone into the careful examination of our study. In order to make a better flow to describe our work and novelty, we check the reference that reviewer #2 mentioned. By doing so, we clarify that **the previously reported work is nothing related to the super-resolved speckle patterns. The established super-resolution methods are based on either fluctuation-based microscopy, ESI, or localization-based microscopy, dSTORM. The Supplementary Figure S11 of the paper shows not the presence of super-resolved speckle but a dSTORM image of multi-mode interference**

pattern of a strip waveguide which visually looks like speckle due to the sparse nature of STORM reconstruction. The interference fringe size of ~ 140 nm corresponds to the diffraction-limit of $\lambda/2n_{\text{eff}}$ within the waveguide. This interference pattern was not used as “a speckle illumination pattern” in their imaging methodology. Moreover, we would like to emphasize that, a most recently published paper from the same group (“Structured illumination microscopy using a photonic chip” (Helle, Ø. I. *et al. Nat. Photon.* (2020)) reports 2.3 times enhancement (~ 209 nm) in imaging spatial resolution by using a photonic waveguide structure to deliver the illumination patterns. The photonic waveguide can generate higher resolution illumination patterns compared to patterns generated in free space purely because of the high refractive index of the “dielectric materials” composing the photonic waveguide (refractive index of Ta_2O_5 or Si_3N_4). Therefore, it is necessary to introduce plasmonic materials/metamaterials for better resolution improvement. To the best of our knowledge, our study is the first experimental demonstration of super-resolution imaging with high- k ($\sim 7k_0$, where k_0 is the wavevector of incident beam) speckle illumination generated by metamaterials.

Supplementary Figure 11 | The multi-mode interference pattern of a strip waveguide imaged using dSTORM. A surface of Alexa 647 dye molecules was prepared on top of the waveguide and a dSTORM image was acquired giving the instant lateral field-distribution of the evanescent field. Interference fringes show structures down to 140 nm as seen in the line profiles. (Scale bar, 5 μm .)

Supplementary Figure 11 from the reference: Diekmann, R., Helle, Ø., Øie, C. *et al.* Chip-based wide field-of-view nanoscopy. *Nature Photon* 11, 322–328 (2017).

[Question 2-2] Moreover, authors commented that previous work requires specific fluorophore dyes which is also in-correct. For intensity fluctuation method to work, either intrinsic photokinetics of the fluorophores are needed or the fluctuations (modulation) of the illumination light itself is sufficient. Both the cases are well known and are being exploited by the community. In this manuscript, the authors have exploited the **fluctuations caused by the illumination, itself, similar to several previous literatures, some of them copied below and many others exists.**

1. Ventalon, C. & Mertz, J. *Quasi-confocal fluorescence sectioning with dynamic speckle illumination. Opt. Lett.* 30, 3350–3352 (2005).
2. Kim, M., Park, C., Rodriguez, C., Park, Y. & Cho, Y. H. *Superresolution imaging with optical fluctuation using speckle patterns illumination. Sci. Rep.* 5, 16525 (2015).
3. K. Guo, Z. Zhang, S. Jiang, J. Liao, J. Zhong, Y. C. Eldar, and G. Zheng, “13-fold resolution gain through turbid layer via translated unknown speckle illumination,” *Biomed. Opt. Express* 9, 260–274 (2018).

[Answer 2-2] Discussion about novelty

Based on the discussion at [Question 2-1], the three references that reviewer #2 mentioned were all discussed in the previous response letter for the reviewers’ comments. We would like to clarify again that **the previously reported illumination-based methods are using diffraction-limited speckles.**

Ref.	Algorithm	Resolution (nm)	Illumination type	Speckle generation methods
1	RMS of image sequence	Diffraction limited (600)	Dynamic speckle	A rotated diffuser plate
2	S-SOFI	140-500	Flickering from fluorophores	A shifted diffuser
3	Ptychographic algorithm	200	Speckle, large FoV with 0.1 NA	Turbid layer

[Question 2-3] *The key advantage of SIM, is the “structured illumination”, and by knowing the illumination periodicity, an important prior knowledge in SIM, it enables SIM with only 9/15 images for 2D/3D SIM image. If 100 images are needed, intensity fluctuations methods would also perform, which will generate results similar as shown in this manuscript (e.g. previous literature, on MUSICAL where 50 nm resolution has already being documented)*

Agarwal, K., Macháň, R. Multiple signal classification algorithm for super-resolution fluorescence microscopy. Nat Commun 7, 13752 (2016).

On using the speckle pattern for illumination, there is no prior information and thus loss of temporal resolution becomes evident. Author stated about pre-determined illuminations using speckle idea but no scientific information was provided, so it is difficult to foresee how this would work. The study carried out in this manuscript is systematic and is more suitable for publication in Scientific Reports.

[Answer 2-3] Discussion about imaging speed and pre-determined illuminations

We would like to remind the reviewer that the number of images needed for the SIM reconstruction is highly related with the resolution improvement factor compared to the diffraction limit. We use more images in our speckle-MAIN mainly because our resolution is much higher than conventional SIM. The periodicity in traditional SIM is an important prior knowledge but pre-determined speckles represent much more prior information enabling improved image reconstruction with less measurements via compressive sensing. This eventually will lead to higher imaging speed.

We agree that our current demonstration of speckle-MAIN is only focused on high lateral resolution achievement using randomly varying high- k speckle illumination. We believe our following study will provide a complete demonstration for high-speed super-resolution imaging. Since the detailed experiment is by itself a mammoth task, it will be undertaken by us in future studies. However, the idea about how to use pre-determined speckle illumination for high-speed imaging has been illustrated in *Nanoscale*, **9**, 18268–18274 (2017). In fact, this is the major distinction between our speckle-MAIN and MUSICAL. The pre-determined speckle illumination makes the compressive sensing scheme possible which will easily improve the imaging speed by 10 folds and the improvement factor can be much greater if the object is sparse. Intrinsic intensity fluctuation cannot be pre-determined, so that no prior information can be utilized in MUSICAL and there is no way to apply the compressive sensing scheme, limiting their potential speed. Please see more detailed discussions in the above mention reference reported by our group.

Response for Reviewer #3

[Initial statement] *From my point of view the authors have addressed most of my initial concerns and the paper is now publishable in Nature Communications. Before publication, I highly recommend the following changes to be made to the manuscript, though:*

[Reply] We appreciate the reviewer's positive evaluation on our study and valuable comments. The detailed responses are below for each of the comments.

[Question 3-1] *Fig. S5 should be moved to the main text - and possibly be integrated with another, already existing figure in the main text (my suggestion would be to combine it with Fig 4 in the main text). The main reason for recommending this change is that the manuscript will greatly benefit from a true proof-of-concept, which is a comparison of the reconstruction of a sub-wavelength structure with an electron micrograph of the same structure - and this is what Fig. S5 provides. It will get lost in the supplement, though....*

[Answer 3-1] We appreciate the reviewer's valuable suggestions. Following the reviewer #3's excellent suggestion, we made modifications in Fig. 4 in the main text.

[Revision in main text]

Figure 4. Speckle-MAIN super-resolution image of fluorescent beads and Qdot 605. a, Diffraction-limited image of fluorescent beads. b, Reconstructed speckle-MAIN image. c, SEM image. Scale bar:

600 nm. **d-f**, zoomed-in images (b) and (c) of fluorescent beads. Scale bar: 100 nm. **h,i**, Intensity cross-section of images (d,f) along indicated direction. **j,k**, Diffraction-limited image. Objective lens: 50×/0.8 NA. **l,m**, Reconstructed speckle-MAIN image of Qdot 605. Scale bar: 2 μm. Exposure time: 200 ms, frame rate: 1 fps.

[Revision in main text] After reduction to only 80 sub-frames, we still can resolve two particles with ~60 nm center-to-center distance by using an objective with NA = 0.8 (Fig. 4 a-i), indicating the robustness of the speckle-MAIN technology.

Figure 4 j-m represents a super-resolution image of a more complex object at a wide field of view. The object is made by a dense drop-casting of quantum dots emitting at 605 nm.

[Question 3-2] *The references in the supplemental information file need to be changed. Reference 2 is not an appropriate reference for SOFI. Here, the original SOFI paper by Enderlein should be cited:*

Fast, background-free, 3D super-resolution optical fluctuation imaging (SOFI), T. Dertinger, R. Colyer, G. Iyer, S. Weiss, J. Enderlein, Proceedings of the National Academy of Sciences Dec 2009, 106 (52) 22287-22292; DOI: 10.1073/pnas.0907866106

Similarly, ref. 3 is not an appropriate reference for ESI. Here, also, the original article should be cited:

Entropy-Based Super-Resolution Imaging (ESI): From Disorder to Fine Detail, I. Yahiatene, S. Hennig, M. Muller, and T. Huser ACS Photonics 2015, 2, 8, 1049–1056, Publication Date: July 14, 2015 <https://doi.org/10.1021/acsphotonics.5b00307>

[Answer 3-2] We thank the reviewer for the correction. As the reviewer correctly pointed out, we have made revisions to the references in the revised Manuscript.

We thank again for all the reviewers' time and efforts that have gone into the careful examination of our manuscript and for giving valuable and insightful comments. These comments have certainly helped us improve our manuscript in the revised form. We hope the revisions we made in the manuscript and our responses have addressed the review comments, and the manuscript in its revised form is considered suitable for publication in *Nature Communications*.

REVIEWERS' COMMENTS

Reviewer #2 (Remarks to the Author):

Authors have clarified my doubts and have performed extensive work during last two revision. I recommend manuscript to be accepted.